# Sleep Duration, Sleep Quality and Physical Activity, but Not Sedentary Behaviour, Are Associated with Positive Mental Health in a Multi-Ethnic Asian Population: A Cross-Sectional Evaluation

**DOI:** 10.3390/ijerph17228489

**Published:** 2020-11-16

**Authors:** Janhavi Ajit Vaingankar, Falk Müller-Riemenschneider, Anne Hin Yee Chu, Mythily Subramaniam, Linda Wei Lin Tan, Siow Ann Chong, Rob M. van Dam

**Affiliations:** 1Research Division, Institute of Mental Health, 10, Buangkok View, Singapore 539747, Singapore; mythily@imh.com.sg (M.S.); siow_ann_chong@imh.com.sg (S.A.C.); 2Saw Swee Hock School of Public Health, National University of Singapore, 12 Science Drive 2, #10-01, Singapore 117549, Singapore; ephmf@nus.edu.sg (F.M.-R.); anne.chu@u.nus.edu (A.H.Y.C.); linda_tan@nus.edu.sg (L.W.L.T.); rob.van.dam@nus.edu.sg (R.M.v.D.); 3Yong Loo Lin School of Medicine, National University of Singapore, 10 Medical Drive, Singapore 117597, Singapore

**Keywords:** positive mental health instrument, mental well-being, personal growth and autonomy, emotional support, health behaviours

## Abstract

Background: We assessed the associations of sleep, physical activity and sedentary behaviour with positive mental health (PMH) in the multi-ethnic population of Singapore. Methods: The Singapore Health 2 study is a nationally representative cross-sectional survey among residents aged 18–79 years. A PMH instrument was administered to 1925 participants to obtain total PMH and six sub-component scores. Self-rated sleep duration, sleep quality, sedentary behaviour and physical activity were assessed. Multivariable linear regression analyses were conducted. Results: The mean age of the participants was 40.1 (SD 14.3) years. Sleep duration (≥8 h/night: β = 0.17,95% CI: 0.02–0.32; 7–< 8 h/night: β = 0.17,95% CI: 0.03–0.30 versus <6 h/night, *p* = 0.002), sleep quality (very good: β = 0.45,95% CI: 0.29–0.60; fairly good: β = 0.20,95% CI: 0.06–0.33 versus very/fairly bad; p*trend* < 0.001) and physical activity (high: β = 0.19,95% CI: 0.05–0.32; moderate: β = 0.15, 95% CI: 0.03–0.27 versus low; p*trend* < 0.001) were directly associated with total PMH. Sedentary behaviour was not significantly associated with PMH. Sleep duration, sleep quality and physical activity were directly associated with the PMH sub-components ‘emotional support’ and ‘personal growth and autonomy’. Conclusions: Duration and quality of sleep and physical activity were directly associated with PMH in an urban Asian population. These findings support incorporating sleep and physical activity interventions to improve population mental health.

## 1. Introduction

About one third of the world’s population is estimated to have insufficient sleep and physical activity [1,2]. Substantial evidence links poor sleep duration and physical activity to adverse health outcomes [3,4]. Sleep durations of less than 6 h or more than 9 h per night are associated with obesity, cardiovascular diseases (CVD) and other chronic conditions [5,6]. The putative relationship between insufficient sleep and chronic diseases such as CVD may also be mediated by psychological distress [7]. Sleep duration and quality have also been linked to several mental conditions such as anxiety, depression and suicidality [8,9,10]. Likewise, low physical activity has also been associated with mental conditions including mood, anxiety and substance use disorders [8]. In line with these findings, interventions to improve sleep and physical activity have consistently reduced psychological distress and mood disorders [11,12].

Although substantial evidence links sleep and physical activity improvement to mental health benefits, a large body of this work originates from their effects on reducing mental distress. While mental disorders are traditionally characterized from negative and abnormal thoughts, emotions and functioning, mental health is defined as a state of physical, mental and social well-being that is “independent of and more than” just absence or presence of illness [13]. Positive mental health (PMH) includes individuals’ attitudes towards themselves, ability to realise their potential, autonomy and environmental mastery, and encompasses subjective and psychological well-being [14]. It is now an accepted notion that mental illness and wellness are not opposing constructs and that people with mental disorders could have beneficial level of PMH. There is therefore a growing emphasis on looking beyond mental illness and devoting attention to PMH in order to optimise gains from mental health interventions [14].

More recent research in mental well-being is indicative of the relationship between health behaviours and lifestyle, and improvements in mental health. For example, longer sedentary behaviour was associated with worse well-being, whereas longer sleep duration and more physical activity were associated with better mental well-being [15,16,17]. However, some of the earlier studies evaluated mental well-being with quality of life measures [16], in younger populations [17] or in chronic conditions [15]. Moreover, to the best of our knowledge, none of the earlier studies attempted to identify sub-components of PMH that may be responsible for the associations with overall mental well-being. Hence, evidence on sleep, physical activity and sedentary behaviour, often referred as movement behaviours [18,19], in relation to PMH remains limited and there is a need for population-based studies evaluating these associations.

To address these gaps, we used data from a cross-sectional survey of a representative sample of adult men and women in Singapore to examine whether movement behaviours are associated with PMH. Specifically, associations for sleep duration, sleep quality, level of physical activity and sedentary behaviour were investigated. We also evaluated which sub-components of PMH may be responsible for associations with overall PMH.

## 2. Materials and Methods

### 2.1. Setting and Population

Singapore is a developed urban economy in southeast Asia with a multi-ethnic population of 5.6 million comprising 74.4% Chinese, 13.4% Malays, 9% Indians and the rest belong to other ethnic groups.

### 2.2. Survey

The Singapore Health (SH-2) study was conducted upon ethical approval from the National University of Singapore Institutional Review Board (NUS-IRB, reference 13–512, approval dated 9 December 2013). Written informed consent was obtained from all participants and from their parent or legal guardian for those aged below 21 years. The study details are described elsewhere [20,21]. Briefly, the SH-2 study, conducted between April 2014 and March 2015, was a cross-sectional population-wide household survey of nationally representative Singapore residents (Singapore Citizens and Permanent Residents) aged between 18 and 79 years, residing for at least 3 months at the enumerated household addresses. A total of 2690 residents participated in the survey giving a response rate of 35%. Of these, 1925 participants who were English literate and had completed a PMH instrument were included in this analysis, resulting in an overall response rate of 25% for the current study. Most of the survey questionnaires were completed through computer-assisted face-to-face interviewer-administered interviews. The PMH instrument was self-administered by the participants at the end of the survey to address social desirability.

### 2.3. Measures

#### 2.3.1. Sleep Duration and Quality

Duration of sleep per night and subjective sleep quality was self-reported by the participants on the Pittsburgh Sleep Quality Index [22]. Sleep duration was obtained from a single item, “during the past month, how many hours of actual sleep on average did you get at night?”, and categorised into less than 6, 6 to less than 7, 7 to less than 8 and 8 or more hours (h) per night [5]. A small proportion of the sample (1.4%) reported having more than 9 h of sleep per night (which has been associated with poor health outcomes). However, it was determined to have minimal effect on the estimates due to the small proportion and combined with 8 h per night sub-group. Subjective sleep quality was rated into four levels ranging from ‘very good’ to ‘very bad’ based on responses to the item “during the past month, how would you rate your sleep quality overall?”.

#### 2.3.2. Physical Activity and Sedentary Behaviour

The Global Physical Activity Questionnaire (GPAQ) was used to assess the duration and frequency of physical activity in a typical week [23]. The 15-item GPAQ also collects information on sedentary behaviour and physical activity in three domains (work, transport and leisure-time). Physical activity of moderate and vigorous intensity is assessed for each domain and expressed as metabolic equivalent tasks (METs)-minutes per week. From these, total METs were derived and categorised into three levels of PA—low (zero METS), moderate (lower median split MET values: Work: ≤3; Transport: ≤7; Leisure-time: ≤2 h/week) and high (upper median split MET values: Work: >3; Transport: >7; Leisure-time: >2 h/week) [20]. An earlier study in adult Singapore residents that assessed criterion validity of interviewer-administered GPAQ, yielded fair to moderate correlation between moderate-to-vigorous physical activity (MVPA) recorded on GPAQ and accelerometer measurements (Spearman’s correlation coefficient = 0.46) and high test-re-test reliability (intraclass correlation coefficient = 0.79) [20]. Sedentary behaviour (in hours per day) was calculated based on a single-item: “How much time do you usually spend sitting or reclining on a typical day?”.

#### 2.3.3. Positive Mental Health

The PMH instrument is a 47-item self-administered measure comprising six subscales—‘general coping’, ‘emotional support’, ‘spirituality’, ‘interpersonal skills’, ‘personal growth and autonomy’ and ‘global affect’ [24]. Participants rate how much each item describes them in general using a six-point scale ranging from ‘not at all like me’ to ‘exactly like me’. Total PMH and subscale scores are obtained by adding respective item ratings and dividing them by the number of items under each domain. Scores range from 1 to 6, with higher scores indicating better mental health. The instrument has demonstrated high validity (Confirmatory Factor Analysis indices: root mean square error of approximation (RMSEA) = 0.047, comparative fit index (CFI) = 0.958, Tucker–Lewis index (TLI) = 0.95) and reliability (Cronbach’s alpha coefficient = 0.961) [21].

#### 2.3.4. Other Covariates

Information on participants’ age, gender, ethnicity, marital status, educational level and employment status were obtained. Body mass index (BMI) in kg/m^2^ was categorised as per Singapore classification criteria for cardiovascular risks (Underweight: <18.5; Normal/low risk: 18.5–22.9; Overweight/Moderate risk: 23.0–27.4; Obese/High risk: ≥27.5 kg/m^2^) [25]. Current smoking status was self-reported and categorised as daily, occasional, past and never smoker. Alcohol consumption in the past 12 months was categorised as excessive (>4 and >3 drinks per sitting for men and women, respectively), non-excessive (≤4 and ≤3 drinks per sitting for men and women respectively) and none [26]. History of chronic physical conditions was defined as having a history of asthma, cancer, diabetes mellitus, CVD or stroke.

### 2.4. Statistical Analyses

Sample weights were applied to the data to derive weighted estimates based on household enumeration exercise, non-response and post-stratification weights based on the age, gender and ethnicity [21]. There were less than 0.1% missing data for all variables due to computerised data collection, except for BMI where 30% data were missing. These were treated as ‘missing’ in the analysis. Mean and standard deviations (SD) were used to describe continuous variables and counts and percentages for categorical variables. First, multivariable linear regression analyses were conducted to evaluate the relationship between sleep duration (four categories: <6, 6–< 7, 7–< 8 and ≥8 h per night), sleep quality (very/fairly bad, fairly good and very good), level of total physical activity (low, moderate and high) and sedentary behaviour (≥11, 8–11 and <8 h per day) as independent variables and total PMH score as dependent variable. To test for linear trends across categories, ordinal categories of the explanatory variables were included as continuous variables in the multivariable models. We used two multivariable models. Model 1 was adjusted for age (years), gender (male, female) and ethnicity (Chinese, Indian, Malay, Other). Model 2 was adjusted for age, gender, ethnicity, marital status (never married, married, separated/divorced/widowed), education level (primary and below, secondary, post-secondary, university), employment status (employed, unemployed), body mass index (underweight, normal, overweight, obese), any chronic physical condition (yes, no), alcohol consumption in past 12 months (excessive, non-excessive, none) and smoking status (daily, occasional, past, never).

Next, to assess the relationships with the six individual PMH sub-components, separate multivariable linear regression analyses for Models 1 and 2 were conducted with the respective PMH subscale scores as the dependent variables. In these models, sleep duration (in hours per night), physical activity (total METS in hours per week) and sedentary behaviour (hours per day) were used as continuous variables as their original forms, while ordinal sleep quality was treated as continuous parameter (beta coefficients denoting 1 category increment in self-rated quality). We were also interested in investigating the role of age and gender on the associations between PMH and movement behaviours. We used two additional multivariable models. In Models 3 and 4, in additional to the Model 2 main effects, we included interaction terms for age group (≤40 years and >40 years, using sample mean as cut-off) and gender (male and female), respectively. The associations of work, transport and leisure-time physical activity were also investigated in Models 1 and 2. Lastly, to address possible influence of poor sleep quality on the relationship of sleep duration with PMH, multivariable regression analyses (Model 2) were repeated after excluding participants with fairly or very bad sleep quality. Regression coefficients and corresponding 95% confidence intervals (CI) were assessed for all analyses. The significance level was set at 0.05. The statistical analysis was performed using SPSS version 24 Complex Samples.

## 3. Results

Characteristics of the sample are presented in Table 1. Participants’ mean age was 40.1 years. The majority were of Chinese ethnicity (71.1%) and married (59.1%). Among the participants, slightly more than half (52.6%) had less than 7 h of sleep per night (18.0% < 6 h and 32.6% 6–< 7 h), 11.1% had very/fairly bad sleep quality, 28.1% had no MVPA and 10.1% had at least 11 h of sedentary behaviour per day. The mean (SD) of total PMH was 4.61 (0.8).

Table 2 presents the association of sleep duration, sleep quality, physical activity and sedentary behaviour with the total PMH score. In Model 1, sleep duration, sleep quality and physical activity were significantly associated with higher total PMH, whereas sedentary behaviour was not significantly associated with total PHM. Further adjustment for education level, employment status, BMI, history of chronic medical conditions, alcohol consumption and smoking (Model 2) did not substantially change these associations. Sleep quality (β = 0.45; 95% CI 0.29–0.60 for very good vs. very/fairy bad quality), sleep duration (β = 0.17; 95% CI 0.02–0.32 for ≥8 vs. <6 h/night) and physical activity (β = 0.19; 95% CI 0.05–0.32 for high vs. low activity) were associated with total PMH.

Associations of movement behaviours with different sub-components of PMH are presented in Table 3. Sleep duration, sleep quality and physical activity were directly associated with ‘personal growth and autonomy’ and ‘emotional support’ scores. For example, with every one-hour increase in sleep duration and better sleep quality, the ‘emotional support’ scores increased by 0.12 (95% CI: 0.07–0.17) and 0.33 (95% CI: 0.23–0.43), respectively. Likewise, 10 MET hours increment in physical activity per week was associated with an increase of 0.06 (95% CI: 0.01–0.10) in the ‘emotional support’ score. Longer sedentary behaviour was associated with a lower score for ‘personal growth and autonomy’, but not with any of the other PMH sub-scales. The spirituality subscale was not associated with any of the explanatory variables. We did not observe any significant change in the associations between total PMH and movement behaviours after investigating interaction by age group and gender in the Models 3 and 4.

With respect to occupational, transport, or leisure time domains, physical activity was associated with higher PMH although associations were less precise and not all were statistically significant (Appendix A). We also evaluated whether the association between sleep duration and PMH only existed among participants with poor sleep quality. However, it was found to be similar after excluding those with fairly or very bad sleep quality (Appendix A).

## 4. Discussion

In this population-based study among adult multi-ethnic Singapore residents, higher sleep duration, better sleep quality and more moderate to vigorous physical activity were associated with better total PMH. In contrast, more sedentary behaviour was not substantially associated with overall PMH. Sleep duration, sleep quality and physical activity were all directly associated with PMH sub-components ‘personal growth and autonomy’, and ‘emotional support’. In addition, sleep quality and sleep duration were directly associated with ‘global affect’ and sleep quality and physical activity were directly associated with ‘general coping’ and ‘interpersonal skills’.

Results in this general population sample are consistent with observations among Scottish CVD patients showing a strong correlation between physical activity and mental well-being assessed with the Warwick Edinburgh Mental Well-being Scale (WEMWBS) [15]. Likewise, in a large sample of adolescents in the United Kingdom, sleep duration of less than 7 h per night and physical inactivity were associated with lower WEMWBS scores [17]. It is difficult to draw further meaningful comparisons given the paucity of research on the relationship of sleep and physical activity with overall mental health. However, the current results have some similarities with earlier studies that reported associations of sleep and physical activity with a range of aspects related to mental well-being. For example, sleep disturbances were associated with lower subjective well-being [27], poor sleep quality and short sleep duration were associated with lower happiness and subjective well-being [28,29] and high physical activity was associated with higher odds of happiness and life satisfaction [30] and psychological well-being [31].

As opposed to the limited data on effects of adequate sleep and physical activity on PMH, the impact of these behaviours on mood disorders and psychological distress is well documented [8,20]. Strong evidence suggests interventions targeting sleep and physical activity are effective in improving mental health and well-being in the community [11,12]. However, the challenge remains in relation to resource planning in determining the correct outcome definition while implementing and evaluating these interventions; whether it should be reduction in psychological distress or improvement in PMH. While the former targets populations that have mental disorders and essentially follows a treatment model, the later operates on the premise of early prevention of distress and mental health promotion for all [32]. Min et al. highlight that regardless of the illness state, mental health promotion works on building resilience and psychosocial well-being [32]. As such, an approach taken from the perspective of bringing about improvements in PMH reaches a broader population. Dissemination of this message to the general public also needs careful consideration. A possible step in this direction would be to break down the construct into smaller, more focused and interpretable components that could enhance mental health promotion, acceptance and effectiveness. For example, in relation to interventions for improving physical activity, which are now almost integral to mental health promotion, attention to increasing social interaction, confidence and sense of meaning and achievement is recommended in order to improve their effectiveness in community samples [33].

This study provided preliminary leads to prioritising PMH sub-components to bring about pronounced changes to the overall PMH. The relationship between sleep duration and PMH seems to be a result of its association with three sub-components—higher ‘emotional support’, ‘personal growth and autonomy’ and ‘global affect’, whereas increase in sleep quality and physical activity were related to almost all sub-components. Sleep duration and quality of sleep are reflective of the ‘sleep-wake cycle’ that is dependent on the circadian (control of daily rhythms and activity of the body and brain) and homeostatic (sleep regulation) processes [34]. Together, these processes regulate the onset and maintenance of sleep and have a moderating effect on the nervous, endocrine and cardiovascular systems. Chronic short sleep duration of 6 or less hours per night has been consistently associated with neurobehavioral impairments, depression, anxiety and suicidality [35]. Longer sleep duration has been associated with positive affect, environmental mastery, personal growth, positive relations with others, purpose in life and self-acceptance [27,36], all of which have conceptual equivalence with the three PMH sub-components associated with sleep duration in our study. It is proposed that ‘feeling well rested’ due to adequate and better sleep could yield a more energetic self, capable of reaching personal goals and being socially connected with others [27,37]. Our findings, thus, provide support to the favourable association between sleep duration, sleep quality and mental health

Contrary to the often-reported inverse association of physical activity with poor mood [38,39], we did not observe an association between physical activity and ‘global affect’. Physical activity may affect mental health through several biological and psychological mechanisms. Physical activity can induce the release of endogenous opioids and beta-endorphin which can act on the central nervous system and reduce stress hormones, enhance the production of neurotropic agents and, thereby, improve mood and experience of calm [40]. In addition, psychological mechanisms of distraction (diversion from unpleasant thoughts), self-efficacy (developing confidence for overcoming challenges) and social interaction (mutual support) may contribute to the association between physical activity and mental well-being [41]. The findings from our study are reflective of two of these, namely ‘emotional support’ (construct equivalence with mutual support) and ‘personal growth and autonomy’ (construct equivalence with self-efficacy), thus providing a plausible underlying mechanism for the observed associations. In relation to the lack of association observed for ‘global affect’ in our study, two systematic reviews investigating the relationship of physical activity with mood have reported weak to no associations in non-clinical populations [42], citing lower self-efficacy and self-motivation as possible explanations. Clinical populations with more severe depressive symptoms may, thus, allow for larger effect sizes than general population samples.

An unexpected finding from this study was the lack of association between sedentary behaviour and PMH. In previous research, more sedentary behaviour was associated with lower self-esteem, lower life-satisfaction and worse psychological well-being [43]. However, these studies were largely focused on children and adolescents and in conjunction with screen time or obesity. While sedentary behaviour was directly associated with depression and psychological distress in adolescents and older adults [43], it was not associated with psychological distress in our sample in an earlier investigation [20]. It has also been proposed that sedentary behaviour and low physical activity are correlated yet distinct factors affecting mental health outcomes via unique bio-psycho-social pathways. While physical activity exerts its influence directly by reducing stress hormones, improving neurotropic agents and stimulating mood-elevating endorphins [40], the indirect impact of sedentary behaviour may be due to its association with other unhealthy behaviours such as increases in food intake and obesity [44]. In an earlier study in Singapore, television (TV) screen time but not computer use and reading time were associated with worse dietary intakes, higher body mass index and greater insulin resistance [45], suggesting a possible effect of a combination of exposures such as food commercials and concurrent snacking related to TV time. The way sedentary behaviour is spent may thus be of importance in relation to PMH.

Our study has several potential limitations. First, because of the cross-sectional design of our study, the direction of cause and effect cannot be established. In addition, our results may be affected by residual confounding due to unmeasured or imperfectly measured confounders. We adjusted for several important socio-demographic and health-related confounders including smoking, alcohol consumption, BMI and history of chronic conditions. However, information on the type of chronic conditions was limited and data on dietary habits (that are linked to mental well-being and can confound the relationship between other health behaviours and outcomes) were not available. Although the PMH instrument was self-administered by the respondents in privacy, there remains a possibility of socially desirable responses for other variables, for example sedentary behaviour. In addition, recall bias in assessing the sleep, sedentary and physical activity behaviour may have weakened associations with PMH. Future research should consider using objective measures such as accelerometers for the assessment of physical activity, sedentary behaviour and sleep duration. The study response rate was modest and only English-speaking participants were included in the PMH assessment. In comparison with Singapore’s general population aged over 25 years, our sample had under-representation of below secondary (3% versus 27%) and over-representation of secondary (40% versus 18%) level education; however, percentages for other education groups were comparable. It is possible that our results do not apply to those with primary or lower education. Lastly, there are likely ethnic and cultural influences on the study findings that limit their generalizability to non-Asian populations. However, consistency of results with studies conducted in the US and Europe, and the expected similarities between urban communities provide some support for the generalizability of these findings to other settings.

## 5. Conclusions

Our findings indicate that sleep duration, sleep quality and physical activity have a direct association with PMH in an adult urban multi-ethnic population in Asia. The association between longer sleep duration and greater PMH after excluding participants with poor sleep quality suggests an independent effect of sleep duration. Longer sleep duration, better sleep quality and higher physical activity were all associated with better ‘emotional support’ and ‘personal growth and autonomy’, but we observed differences in association for other aspects of PMH. For example, sleep duration was associated with better ‘global affect’, whereas physical activity with better ‘general coping’ and ‘interpersonal skills’. Further research is needed to assess the relationship between sedentary behaviour and PMH. Together, these results add to the growing body of evidence pertaining to the benefits of sleep and physical activity to PMH and provide indication of the likely components of mental health that could be targeted through sleep- and physical activity-based interventions, in order to improve overall mental well-being in the population.

## Figures and Tables

**Table 1 ijerph-17-08489-t001:** Characteristics of the study sample and total positive mental health (PMH) scores for sub-groups (*n* = 1925).

		*n*	%	Total PMH Score	
				Mean	SD
Age (Mean, SD)	40.1, 14.3				
Gender	Men	921	52.1	4.58	0.77
	Women	1004	47.9	4.66	0.76
Ethnicity	Chinese	1149	71.1	4.50	0.73
	Malay	320	14.1	4.93	0.78
	Indian	366	11.0	4.79	0.84
	Other	90	3.9	4.91	0.72
Marital status	Never married	583	34.9	4.50	0.68
	Married	1168	59.1	4.68	0.81
	Separated/Divorced/ Widowed	170	6.1	4.61	0.81
Education level	Primary and below	70	3.0	4.78	1.19
	Secondary	810	40.0	4.64	0.84
	Post-secondary	424	24.0	4.57	0.72
	University	621	33.0	4.60	0.65
Employment status	Employed	1395	73.5	4.62	0.74
	Unemployed (student, homemaker, retired)	527	26.5	4.60	0.85
Alcohol consumption in past 12 months *	Excessive	98	6.0	4.54	0.64
	Non-excessive	861	49.6	4.52	0.72
	None	966	44.4	4.73	0.82
Smoking status	Daily	231	11.5	4.60	0.81
	Occasional	59	3.3	4.58	0.62
	Past	147	7.5	4.65	0.76
	Never	1488	77.7	4.61	0.76
Any chronic physical condition	Yes	815	40.6	4.63	0.82
	No	1110	59.4	4.60	0.73
Body Mass Index	Underweight	69	6.0	4.35	0.68
	Normal	407	33.4	4.52	0.72
	Overweight	514	39.5	4.65	0.76
	Obese	301	21.1	4.71	0.79

* Categorised as Excessive if consumed >4 and >3 drinks per sitting for men and women, respectively, Non-excessive if men had ≤4 and women had ≤3 drinks per sitting and None for no consumption.

**Table 2 ijerph-17-08489-t002:** Associations of sleep and physical activity related behaviours with the total positive mental health score.

			Multivariable Model 1 *				Multivariable Model 2 ^#^			
			β	95% CI		*p* Trend	β	95% CI		*p* Trend
	*n*	%		Lower	Upper			Lower	Upper	
**Sleep duration (hours per night)**						0.002				0.002
≥8	623	32.6	0.147	0.025	0.269		0.171	0.022	0.321	
7–< 8	556	29.3	0.150	0.039	0.261		0.165	0.029	0.302	
6–< 7	372	20.1	0.009	−0.106	0.124		−0.009	−0.150	0.132	
<6	374	18.0	Ref				Ref			
**Sleep quality**						<0.001				<0.001
Very good	1283	67.1	0.459	0.330	0.587		0.446	0.291	0.601	
Fairly good	424	21.8	0.253	0.149	0.358		0.196	0.063	0.330	
Very/fairly bad	218	11.1	Ref				Ref			
**Physical activity ** ^@^						<0.001				<0.001
High	767	38.3	0.204	0.094	0.315		0.188	0.051	0.324	
Moderate	614	33.6	0.122	0.022	0.221		0.147	0.025	0.268	
Low	502	28.1	Ref				Ref			
**Sedentary time (hours per day)**						0.176				0.142
<8	617	34.4	0.017	−0.117	0.152		0.086	−0.086	0.258	
8–11	1130	55.5	−0.026	−0.163	0.110		0.043	−0.131	0.216	
≥11	178	10.1	Ref				Ref			

* General linear regression models adjusted for: Age, Gender (Men, Women), Ethnicity (Chinese, Malay, Indian, Other). ^#^ General linear regression models adjusted for: Age, Gender (Men, Women), Ethnicity (Chinese, Malay, Indian, Other), Marital status (Never married, Married, Separated/Divorced/Widowed), Education level (Primary and below, Secondary, Post-secondary, University), Employment status (Employed, Unemployed), Body mass index (underweight, normal weight, overweight, obese), Any chronic physical condition (Yes, No), Alcohol consumption in past 12 months (Excessive, Non-excessive, None), Current smoking status (Daily, Occasional, Past, Never).^@^ Categorised from total metabolic equivalent tasks (METs) in minutes per week, from moderate to vigorous levels of activity in work, leisure and transport domains; categorised into three levels: low (zero METS), moderate (lower median split MET values) and high (upper median split MET values).

**Table 3 ijerph-17-08489-t003:** Associations of sleep duration, sleep quality, physical activity and sedentary time with different domains of positive mental health.

	Multivariable			
	β	95% CI		*p* trend
		Lower	Upper	
**Sleep duration (hours per night)**				
Total Positive Mental Health	0.058	0.022	0.094	0.002
General coping	0.037	−0.010	0.083	0.120
Emotional support	0.118	0.068	0.169	<0.001
Spirituality	0.048	−0.032	0.127	0.236
Interpersonal skills	0.025	−0.013	0.063	0.201
Personal growth and autonomy	0.047	0.004	0.089	0.030
Global affect	0.108	0.064	0.153	<0.001
**Sleep quality ^#^ (1 category increment in rated quality)**				
Total Positive Mental Health	0.204	0.135	0.273	<0.001
General coping	0.193	0.095	0.291	<0.001
Emotional support	0.330	0.231	0.428	<0.001
Spirituality	0.084	−0.079	0.246	0.312
Interpersonal skills	0.135	0.057	0.214	0.001
Personal growth and autonomy	0.185	0.101	0.269	<0.001
Global affect	0.389	0.304	0.473	<0.001
**Physical activity (Total METS in hours per** **week, estimates expressed for 10 MET hours/week)**				
Total Positive Mental Health	0.060	0.030	0.090	<0.001
General coping	0.080	0.030	0.120	<0.001
Emotional support	0.060	0.010	0.100	0.009
Spirituality	0.040	−0.040	0.110	0.311
Interpersonal skills	0.070	0.030	0.110	<0.001
Personal growth and autonomy	0.080	0.040	0.120	<0.001
Global affect	0.030	−0.010	0.070	0.209
**Sedentary time (hours per day)**				
Total Positive Mental Health	−0.012	−0.027	0.004	0.142
General coping	−0.010	−0.032	0.011	0.346
Emotional support	0.000	−0.020	0.020	0.985
Spirituality	−0.015	−0.047	0.018	0.374
Interpersonal skills	−0.010	−0.026	0.007	0.255
Personal growth and autonomy	−0.023	−0.040	−0.006	0.008
Global affect	−0.007	−0.025	0.011	0.420

^#^ Sleep quality categorised as 1: Very bad, 2: Fairly bad, 3: Fairly good and 4: Very good.

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
