# Peer review of "Sleep Duration, Sleep Quality and Physical Activity, but Not Sedentary Behaviour, Are Associated with Positive Mental Health in a Multi-Ethnic Asian Population: A Cross-Sectional Evaluation"

_ijerph, 2020, doi:10.3390/ijerph17228489_

Round 1

Reviewer 1 Report

Overall it is logical and well documented. However, many studies have already been conducted using accelerometers that can measure physical activity or sleep more objectively. And as the authors said, it is difficult to confirm cause-effects, which is a limitation of the study. It is well known that mentally healthy people can sleep better and be more active in physical activity, and vice versa. 

The survey was conducted in a fairly wide range of ages from 18 to 79 years, but it would be good to conduct further analysis to see if the results were different for each age group. Also, the average age of responders was presented, but I am curious that the response rate may have been different for each age group.

Line 94

Please revised the error. :  8 or more hours per night5.

Line 136

 Please check four categories of sleep duration: (four categories: <6, 6-<7, 7-<8 and ≥8 hours per night)

Line 144-145

 Please describe the cut off value of the BMI categories: underweight, normal, overweight, obese

Author Response

Dear Reviewer 1,

Thank you for reviewing our manuscript and your helpful suggestions. Our response to these can be found in the attached document.

We hope these are acceptable.

Thank you.

Regards,

Janhavi

Reviewer 2 Report

The authors in this manuscript investigated the effects of sleep, physical activity, and sedentary behavior, on positive mental health (PMH). This study is quite relevant and well-designed. I have only few minor concerns as follows"

  • Sleep varies with age and gender. Did authors find any age/gender based difference in determining the effects of sleep on PMH. If so, please describe that in the discussion.
  • In the sentence "evidence on sleep, physical activity, and sedentary behaviour, often referred as movement behaviours," the authors described sleep as movement behavior which is incorrect. Either rephrase the sentence or provide one or two references.
  • In Table 2 and 3, P should be in small letter.

Author Response

Dear Reviewer 2,

Thank you for reviewing our manuscript and your helpful suggestions. Our response to these can be found in the attached document.

We hope these are acceptable.

Thank you.

Regards,

Janhavi

Reviewer 3 Report

Dear authors, thank you for your efforts with this research. It piqued my interest and provides an valuable insight on the links between movement behaviour and positive mental health. Please find below my comments:

Line 45-50 I appreciate the discussion that is framed on the positive state of mental health.

Line 94 "night5", I assume this is an error.

Line 106-107 Please indicate the MET values for these categories.

Line 120 Please include the coefficients for validity and reliability for the PMH.

Line 124 Please include the BMI classification criteria for the benefit of readers.

Line 162-163 Results show only marginally more than half (52.6%) had less than 7 hours. Suggest rephrasing to indicate so rather than "most".

Line 176-177 Adding ">8 hr vs. <6 hrs" and "high vs. low" for sleep duration and physical activity respectively. 

Line 193-196 I struggled to understand the reporting of these specific results. Could the presentation of these result be worded more clearly?

Line 239-245 I appreciate the mention of these public health recommendations.

Line 295 Here is a great research opportunity to investigate this in a sub sample of the current cohort.

Line 305 I don't believe it is "measurement error" per se but rather the fact these data are self-reported measures open to recall and relativity of estimates.

This research provides an impetus for more objective and validated studies. With the ubiquitous use of consumer-based activity trackers, and sleep apps, by the general population to self-monitor and change health behaviours, there is potential to further enhance the research with a sub sample of participants, tracking their movement behaviours in free-living context. The collective findings can help inform the development of national movement guidelines.

Author Response

Dear Reviewer 3,

Thank you for reviewing our manuscript and your helpful suggestions. Our response to these can be found in the attached document.

We hope these are acceptable.

Thank you.

Regards,

Janhavi

Round 2

Reviewer 1 Report

The revised manuscript is well described as the comments of the round 1 review. Thank you for constructing the research with a considerable number of participants.